# Substantiating the Therapeutic Effects of Simultaneous Heat Massage Combined with Conventional Physical Therapy for Treatment of Lower Back Pain: A Randomized Controlled Feasibility Trial

**DOI:** 10.3390/healthcare11070991

**Published:** 2023-03-30

**Authors:** Tae-Hwan Kim, Soon-Kwon Park, Il-Young Cho, Jong-Hoo Lee, Hong-Young Jang, Yong-Soon Yoon

**Affiliations:** 1Department of Rehabilitation Medicine, Presbyterian (Jesus) Medical Center, Jeonju 54987, Republic of Korea; 2College of Social Sciences, Jeonju University, Jeonju 55069, Republic of Korea; 3College of Medical Sciences, Jeonju University, Jeonju 54896, Republic of Korea; 4Research Center of Welllife Healthcare, Sungkyul University, Anyang 14097, Republic of Korea

**Keywords:** lower back pain (LBP), heat massage, physical therapy

## Abstract

Background: There are various therapeutic options for the conservative management of lower back pain (LBP). A combination of two or more treatment options may be more effective in the clinical management of non-specific LBP. In this study, we compared the effects of simultaneous heat massage with conventional physical therapy in patients with subacute LBP. Methods: A single-center randomized controlled trial in which 40 participants with LBP were allocated to one of two groups: a heat massage group (HMG) and physical therapy group (PTG). The HMG received simultaneous heat massage therapy using a mechanical device (CGM MB-1401, Ceragem, Republic of Korea). The PTG received conventional physical therapy. Both groups received 40 min of therapy once daily, five times a week, for a total of four weeks. Changes in serum cortisol, epinephrine (EP), and norepinephrine (NE) were assessed. The outcomes were measured using the pain numeric rating scale (PNRS), the Oswestry disability index (ODI), the Roland–Morris disability questionnaire (RMDQ), the short-form McGill pain questionnaire (SF-MPQ), the multidimensional fatigue inventory (MFI-20), the Beck depression inventory (BDI), surface EMG (sEMG), and sympathetic skin response (SSR) at baseline (PRE), at 2 (2 W) and 4 weeks (4 W) following the intervention. Results: The serum EP and NE levels in the HMG decreased after treatment. The PNRS, ODI, RMDQ, and SF-MPQ scores improved without significance in both groups. The BDI score showed improvement in the HMG before the PTG. The MFI-20 score improved in both groups, but the results were better in the HMG than in the PTG at 4 W. All the activities of sEMG were significantly decreased in both groups. However, the improvement of the %MVIC in the HMG was better than that in the PTG at 4 W. The SSR latency on sEMG decreased while the amplitude increased in the HMG at 2 W and 4 W, respectively. Conclusions: Following 4 weeks of combined therapies, heat massage was not superior to conventional physical therapy alone. Both treatments were shown to be effective in improving LBP and pain-related disability. However, heat massage was shown to have a better effect on the control of autonomic nerve function and underlying moods.

## 1. Introduction

Lower back pain (LBP) is a widespread problem in clinical practice [1]. Research studies have reported that LBP compromises function and causes more disability than most other health conditions worldwide [2]. The functionality of individuals is profoundly impacted. It affects our performance in activities in daily living, including leisure, and affects our quality of life by preventing us from fulfilling our duties and responsibilities at work [2]. The literature indicates that approximately 60–80% of the world’s population will experience an episode of LBP at some point, and among these individuals, 60–85% are expected to have more than one episode in their lifetime [1,2].

The clinical management of LBP broadly ranges from conservative to intensive approaches and includes surgical interventions in some cases [3]. A non-surgical approach usually involves a regimen of medication and rest, as well as therapeutic modalities, such as exercise, stretching, heat, and massage. The dual purpose and primary goal in the conservative management of LBP are to preserve the baseline function and to prevent disability [3]. Thus, many will opt for efficient and conservative treatment methods that ensure an uninterrupted lifestyle with a speedy return to work and continued community participation.

The most common conservative treatment approach to the management of non-specific pain is the application of heat and low-frequency TENS. The clinical effects of thermal therapy produce temporary pain relief from enhanced blood flow and metabolism and improve the elasticity of the connective tissues [4]. A heat wrap is more effective, and one study compared it with pharmacotherapeutics—acetaminophen or ibuprofen—for pain relief and showed improved scores on the Roland–Morris disability questionnaire (RMDQ) [5].

On the opposite end of the spectrum, in terms of pain management, massage therapy is considered to be one of the oldest and most widely used non-pharmacologic approaches to LBP management. It promotes circulation and increases lymphatic drainage, improves the elastic and inelastic properties of connective tissues and muscles, reduces muscle pain, relaxes rigid muscles, and systemically relaxes the body [6]. In a previous Cochrane study, massage was determined to be an effective treatment option for the treatment of acute, subacute, and chronic LBP [7]. Massage therapy, such as heat therapy, has been shown to be more effective in the treatment and management of subacute LBP when combined with other therapeutic approaches than medication, exercise, or education alone [8].

The clinical effects of combined heat and massage application include increased blood flow to the affected areas, reduced muscle spasms, and increased flexibility [9]. If heat and massage are applied at the same time, these effects are expected to be intensified. Previous studies have shown that simultaneously combining two or more treatments for LBP is more effective than single-modality management [8,10]. Heat plus exercise provides greater pain relief with improved RMDQ scores when compared with exercise alone in LBP patients [10]. The combined thermal massage is presumed to have a synergistic effect. However, there have been no studies comparing the effects of the direct application of simultaneously combined heat and massage against conventional therapy for pain relief. This knowledge gap serves as the rationale for a clinical investigation using a therapeutic thermal massage device. Thus, the purpose of this study was to investigate the clinical improvement of pain and function by comparing the treatment effects of heat-structured massage and conventional physical therapy in individuals with LBP.

## 2. Materials and Methods

### 2.1. Design and Sample Size

This pilot study is a single-blinded, prospective, randomized, controlled trial. The study was approved by the Institutional Review Board of the Presbyterian Medical Center (IRBN. 2019-03-006) and was registered with the Clinical Research Information Service (KCT0004654; 28 January 2020). All ethical considerations were addressed and incorporated in the explanation of the research study to participants, and their written consent was obtained thereafter. We enrolled 20 participants in one of two groups (*n* = 40) through convenience sampling and exceeded the minimum recommended number of participants for pilot studies [11].

### 2.2. Participants

Randomization was achieved using a centrally generated, variable-sized block design created by a statistical program that was concealed and protected. Physicians and collaborating researchers were blinded to the type of management. The participants knew if they received heat massage or physical therapy, but they were blinded to the way in which they received the treatment. The study was conducted by a third-party clinical research coordinator.

Forty-six patients with nonspecific LBP were recruited from an outpatient clinic through a 3-week poster advertisement period. The enrollees were randomized into either physical therapy (PTG) or heat massage treatment (HMG) groups based on the random allocation sequence. The two groups were managed in separate buildings of the medical center to ensure blinding. The clinical characteristics of the participants, including age, body weight, height, BMI, and history, were recorded before group allocation.

Patients were eligible for enrollment if they were aged 18 years and over, experienced LBP for more than 6 weeks, had a Roland–Morris disability questionnaire (RMDQ) score of 4–6 points, and an Oswestry disability index (ODI) score of 40% or lower. Patients were excluded from the study if they had a history of recent illness, malignancy, pregnancy, muscle disease, surgical history of the lumbar or thoracic back region, or inflammatory spine disease. They were excluded if they were taking medications, including non-steroidal anti-inflammatory drugs (NSAIDs); if they had cognitive dysfunction, compression of the dorsal root, or discogenic pain. Patients who scored 7 points or higher on the pain numeric rating scale (PNRS) with accompanying pain radiating to the lower limbs were also excluded, as were those experiencing muscular weakness and abnormal sensations in the lower limbs. Finally, patients with positive RA, ANA, HLA-B29, and CRP through laboratory testing were also excluded before the start of the clinical trial.

### 2.3. Intervention

The heat massage device (CGM MB-1401; Ceragem, Seoul, Republic of Korea) was run in the eighth mode (this mode provides acupressure, moxibustion, and rubbing along the spine) in the HMG study (Figure 1). Once the participant laid down on the device in the supine position, a rolling projector protruded up against the posterior aspect of the spinal column to scan its curvature along the sagittal plane from the cervical spine through to the sacrum. These rollers applied heat and massage while automatically adjusting the height of the protrusion with reference to the curvature of the entire length of the spine. All participants used the device for 40 min, once daily, 5 days per week, for a total of 4 weeks.

In the PTG, the participants received physical therapy treatment (from 4 physiotherapists) for up to 5 days per week for 4 weeks in total. Treatment consisted of a typical physiotherapy program without exercises, with the application of 5 min of therapeutic ultrasound (US), 10 min of transcutaneous electrical nerve stimulation (TENS), and 15 min of hot pack application. TENS was applied with a frequency of 75~125 Hz and an intensity of 40 mA. Ultrasound was applied for 5 min with a frequency of 1MHz and an intensity of 1.5 W/cm^2^. Both treatments were applied to specific areas of the lower back where patients complained of pain. This is the representative non-surgical approach to the conservative therapeutic regimen commonly used for LBP patients in the Republic of Korea [12].

Participants who received physical therapy or heat massage for at least one session were reported to the researcher if they experienced any adverse events.

### 2.4. Measurement Tools

The pain numeric rating scale (PNRS), the Oswestry disability index (ODI), and the Roland–Morris disability questionnaire (RMDQ) were used as tools to assess the back pain intensity and its impact on pain and function [13]. The PNRS is a unidimensional measure of pain intensity and is a segmented numeric version of the visual analog scale (VAS), in which a participant selects a whole number (0–10 integers) that best reflects the intensity of pain [14]. The ODI is used to measure the LBP-related dysfunction and consists of 10 questions—5 points for each question—and the total score is expressed as a percentage. We used the validated Korean version of the ODI in this study [15,16]. The RMDQ is a tool consisting of 24 questions—scored from 0 to 24—and has been reported to closely reflect the changes over time in patients with LBP. It was designed for use in research to assess physical disability due to LBP and is more discriminative among patients who have relatively little disability over those who have a high level of disability [15,17].

The short-form McGill pain questionnaire (SF-MPQ) is an abbreviated version of the widely used MPQ and is a combination of a descriptive scale, VAS, and present pain intensity (PPI) scale. The descriptive scale and pain rating index are based on 15 selected words, including a sensory subscale with 11 words and an affective subscale with 4 words [18].

The Beck depression inventory (BDI) and the multidimensional fatigue inventory (MFI-20) were used as parameters to assess pain-related depression and fatigue.

Surface electromyography (sEMG) and the sympathetic skin response (SSR) were measured to analyze the voltage level of the spinal muscles and the changes in sympathetic activities.

Using the FREEEMG device (BTS Bioengineering Corp, Boston, MA, USA), a sampling rate of 1000 Hz and a band-pass filtered from 20 to 500 Hz, the sEMG signals of the spinal extensor muscles were obtained. This examination was performed at rest and in the most flexed position. The pairs of electrodes were placed on the right and left sides of the muscles. Upon finding the L3 level through palpation, each of the two recording electrodes was attached at the L3 vertebral level, 2–3 cm away from the center of the spine. The reference electrode was attached 10 cm above the L3 recording electrode [19].

In order to assess activities during the motion of trunk flexion extension, sEMG was measured in the lumbar paraspinal muscles. The participant was in an upright standing position with arms crossed over their chest, then flexed about the waist fully forward, and held this position for 6 s before returning to a straight standing position. The data obtained for 6 s in a completely flexed posture showed an average RMS value of 4 s, excluding 1 s before and after. Participants were placed in the prone position for the isometric contraction test with the lower limbs fixed to the bed with straps, and the MVIC (maximum voluntary isometric contraction) was measured for 6 s while the lumbar extensors were maintained in the prone position. The acquired data, treated as the RMS with the average value for 4 s, excluding the first and last second, was recorded as the MVIC [20,21]. All tests were performed 3 times to obtain an average value. To compare the EMG activity of the MVIC and the full flexion posture, the sEMG signals were normalized by dividing the RMS (%MVIC) [22,23].

The sympathetic skin response (SSR) was monitored using a Viking IV (CareFusion, San Diego, CA, USA) instrument to measure changes in the sympathetic nervous system reflecting pain-induced stress. The filter settings were adjusted (0.2~20 Hz), and the participant was laid in the supine position in a quiet, dimly lit room with an ambient temperature between 22 and 24 °C. A round surface electrode with a diameter of 1 cm was attached 2 cm from the medial border of the left palm, while a reference electrode was attached to the dorsal surface of the right hand. Nerve stimulation was applied 3 times on the proximal part of the volar wrist on the right medial nerve distribution surface, and the intensity was set at 15~20 mA.

All parameters, except for blood sampling, were evaluated before the start of the clinical trial (PRE), after 2 weeks (2 W), and at 4 weeks (4 W). The laboratory tests included evaluation of the epinephrine (EP), norepinephrine (NE), and cortisol levels before the clinical trial (PRE), with the changes in these levels compared at 4 weeks (4 W). Coffee, tea, chocolate, and allergy medications were prohibited a day before blood sample collection.

### 2.5. Statistics

All statistical tests and analyses were conducted using SPSS for Windows (Version 22.0; SPSS, Inc., Chicago, IL, USA). A *p*-value of <0.05 was considered statistically significant. A Shapiro–Wilk test was performed to assess the normality of the distribution. The independent sample t-test, Mann–Whitney U-test, and Chi-squared test were conducted to determine the general characteristics and homogeneity of the pre-measured values in both groups. In order to verify the effectiveness of the thermal massage device, the independent sample t-test and Mann–Whitney U-test were used to compare the outcomes between the two groups. Repeated measures of ANOVA and Friedman test with a Bonferroni correction assessed the between-group differences regarding changes in the PNRS, SF-MPQ, RMDQ, and ODI.

## 3. Results

### 3.1. General Characteristics

A total of 40 participants were evaluated after the exclusion of six due to their unwillingness to continue with the study. There were no significant differences between the experimental and control groups in the general characteristics (Table 1). The absence of significant differences (*p* > 0.05) between the two groups on homogeneity testing was deemed suitable for conducting the trial (Table 2).

### 3.2. Blood Chemistry

Cortisol levels were not different before and after treatment, and these values were not significantly different between the two groups, as shown in Table 3. However, there was a significant decrease in serum EP and NE levels at 4 weeks compared to baseline in the HMG (*p* < 0.05). This is summarized in Table 2.

### 3.3. Pain and Pain-Related Disability

The PNRS, ODI, RMDQ, and SF-MPQ scores improved after 2 W and 4 W in both groups when compared to before the trial. There was no significant difference after treatment between the HMG and PTG (*p* < 0.05) (Table 4 and Table 5).

### 3.4. BDI and MFI-20

In the HMG, the MFI-20 was gradually improved to PRE (57.50 ± 2.07), 2 W (54.44 ± 4.06), and 4 W (47.50 ± 5.38) with treatment. Similarly, in the PTG, the MFI-20 was improved to PRE (58.17 ± 2.18), 2 W (54.28 ± 3.80), and 4 W (51.22 ± 3.75). The MFI-20 of both groups were improved at 2 W, but there was no significant difference. However, at 4 W, The MFI-20 of the HMG showed a significant improvement compared to that of the PTG (*p* < 0.05) (Figure 2).

There was a gradual improvement in the BDI with treatment in both groups. In the HMG, the BDI was improved to PRE (10.83 ± 3.70), 2 W (9.44 ± 3.52), and 4 W (7.61 ± 1.9). In the PTG, the BDI was improved to PRE (11.11 ± 3.18), 2 W (10.11 ± 3.32), and 4 W (9.39 ± 4.10). The BDI of the PTG was improved at 4 W compared to PRE and 2 W, whereas that of the HMG was improved at 2 W (*p* < 0.05) (Figure 2).

### 3.5. EMG and SSR

The levels of the normalized sEMG signal (%MVIC) were changed at PRE (60.17 ± 20.67), at 2 W (59.35 ± 16.97), and at 4 W (42.94 ± 11.22) among the HMG based on treatment progress; changes were also observed at PRE (59.58 ± 19.92), at 2 W (56.02 ± 18.80), and at 4 W (54.15 ± 21.17) in the PTG. The %MVIC was significantly decreased at 4 W in both groups. However, there was a decrease in the HMG compared to the PTG at 4 W (*p* < 0.05) (Figure 3).

The SSR measurement for the HMG showed a decreased latency with increased amplitude at 4 W and at 2 W. However, for the PTG, the latency decreased, and amplitude increased at 4 W and at 2 W, respectively, with similar results for PRE. In comparison between the two groups, the latency, and amplitude of the SSR were significantly different at 2 W and 4 W compared to at PRE (*p* < 0.05) (Figure 3).

### 3.6. Safety

There were no adverse events observed in either group during each visit.

## 4. Discussion

The study aimed to verify changes in the level of disability and symptoms related to LBP by undergoing physical therapy or heat massage treatment options among patients who have had underlying back pain for more than 6 weeks.

The literature shows that the levels of epinephrine (EP) and norepinephrine (NE) are higher in patients with frequent episodes of back pain; labor-related stress factors were associated with inducing musculoskeletal pain and NE concentration [24,25]. In our study, we observed a significant decrease in serum EP and NE levels at 4 W among members of the HMG. The serum levels of EP and NE serve as indicators of the sympathetic nervous system (SNS) function under a normal physiologic state. The decrease in these levels is reflective of the decreased function of the sympathetic nerve input, presumably as a direct result of the heat massage application. However, the measurement of catecholamines can be influenced by factors such as time, diet, tobacco, caffeine, position, exercise, and stress, as well as by techniques in phlebotomy and specimen collection. Therefore, further research is necessary to elucidate the precise role of these indicators in establishing pain.

In this study, the *ODI* and *RMDQ* evaluated the functional level of daily life, and the *SF-MPQ* was used to assess the degree of pain and related emotional states among those with LBP. These are specific measurement tools related to LBP rather than an indication of the patient’s performance, disability, and overall health. Although both groups showed improvement in LBP after the intervention, there were no differences between the two groups regarding pain and pain-related indicators, such as the VAS, ODI, and RMDQ. The results suggest that both clinic-based PT with simultaneous heat massage application can effectively control LBP and function.

The *BDI* and *MFI-20* were assessed to observe the changes in the quality of life based on pain improvement by comparing both types of management, and these were shown to be effective using these measurements. Fatigue and accompanying various morbidities, including LBP, can also profoundly impact the quality of life. Participants with LBP and fatigue had greater pain intensity, more depressive symptoms, and increased levels of disability compared to those without substantial fatigue. Further, chronic LBP patients were found to be more fatigued [26,27].

Pain alone can be a single cause of fatigue. Fatigue and pain have a synergic effect in reducing the functional capacity in LBP patients. The loss of function may be increased due to the simultaneous occurrence of fatigue and LBP [28]. Therefore, fatigue should be simultaneously assessed when evaluating pain in patients with LBP. The MFI-20 is a self-reported instrument designed for the measurement of fatigue. Although there have been no studies addressing fatigue using massage application in LBP with the MFI-20, a previous study of cancer patients reported that massage therapy produced significant relief for fatigue in this patient group [29].

Previous studies have shown higher scores for depression among patients with acute or subacute non-specific LBP than among asymptomatic participants without underlying pain [30], and depression may affect the prognosis of acute and subacute LBP, according to a systematic review [31]. In this study, the first improvement of depression in the HMG was observed at two weeks. It is plausible that this may be attributed to the effects of massage therapy. A previous study showed that massage is effective for clinical depression [32]. In a study on the progression of depression associated with LBP, it was reported that patients who experienced substantial improvement in LBP showed simultaneously improved levels of depression. Hence, in the treatment of LBP, it is advantageous to evaluate the degree of pain and underlying depression [33].

In the meta-analytic review for sEMG, as an objective marker of LBP, sEMG is a simple and noninvasive measure of muscle activity [23]. The normalization of sEMG is performed when comparing *EMG* amplitudes between individual muscle fascicles or between subjects because the RMS value of sEMG is affected by the age, sex, and body fat of the individual. For example, the amplitudes of sEMG will be decreased in those with higher fat and less water content in the body [34]. In this study, there were no differences in age, sex, and BMI in both groups prior to intervention, and the BMI was shown to be appropriately proportional to body fat [35]. The most common method of normalizing EMG signals is to use the maximal voluntary isometric contraction (MVIC) method as the reference value [22]. 

The flexion-relaxation phenomenon (FRP) is defined as the reduced EMG activity of the lumbar erector spinal muscle during full trunk flexion, and it is achieved in healthy individuals but not in most LBP patients. The FRP is the most studied sEMG response among the various physiologic indicators of LBP [23,36]. The levels of sEMG activities after normalization (%MVIC) were significantly decreased in both groups. However, at 4 W, the %MVIC of the HMG was decreased, while the PTG showed no changes. It is plausible that both groups had a decrease in sEMG signals at full flexion due to an improvement in LBP. The improved muscle relaxation among the HMG is presumably the result of an additive effect of the heat combined with deep massage. However, further research is warranted to elucidate the precise pressure measurement applied by the massage device.

Musculoskeletal pain triggers constant stress, which activates the SNS, resulting in increased blood pressure, tachycardia, dizziness, anxiety, and muscle tension. Persistent pain is accompanied by physiologic changes within the autonomic nervous system (ANS) [37]. The excessive stimulation of the SNS may contribute to the intensity and chronicity of pain via the increased nociceptor response [38,39]. When functional changes occur due to stress in the ANS, an electrodiagnostic examination reveals the presence of *SSR*, usually with increased amplitude and decreased latency [40]. 

In our measurement of the *SSR*, the latency increased while the amplitude decreased in both groups. However, in the HMG, the latency and amplitude of the SSR after treatment were different compared to those of the PTG. A previous study concurrently showed that the application of simultaneous thermal massage reduced sympathetic activity, and similar results have also been observed in the SSR [41]. These observations are presumably attributed to a decrease in sympathetic activity with associated catecholamine changes following the thermal massage.

A Cochrane review for LBP concluded that massage might be beneficial for subacute and chronic nonspecific LBP, especially when combined with exercise and patient education [42].

Massage therapy benefits the human body through muscular relaxation, improved blood and lymph circulation, and neurohormonal immunologic effects. Moreover, when heat is combined with massage and is applied to a painful area, the resultant benefits are further heightened. It promotes increased blood flow with vasodilatation, an increased metabolic rate, relaxation of muscle spasms, pain relief via the gate-control mechanism with reduced ischemia, and increased connective tissue elasticity. Circulatory vessels also widen to enhance blood flow to transport excess lactic acid and other toxins away from fatigued muscles. It also relaxes muscle tone by reducing muscle spindle and gamma-efferent firing rates [43]. A Cochrane review for LBP concluded that massage might be beneficial for subacute and chronic nonspecific LBP, especially when combined with exercise and patient education [42].

Therefore, when applying both heat and massage simultaneously, it was expected that a synergistic effect of the two applications would have a better impact than conventional physical therapy. Ironically, the results from this study did not confirm this expectation. On the contrary, in a previous study on the effects of massage on LBP, the results showed that massage was significantly better than general physical therapy in improving pain or disability [44].

Other studies have also shown that hand massage is therapeutically better than physical therapy modalities for the management and decrease of (1) pain intensity, (2) disability, and (3) missed work days [45,46].

There is the likelihood of potential confounding bias in terms of the therapist–patient relationship, as well as the techniques and experiences in the application of therapeutic hand massage. These confounding variables should be controlled in order to obtain a true effect size as much as possible. To control for this and standardize the delivery of intervention in our study, the massage device was used. In this study, the comparative analysis between heat massage and physical therapy and the resultant effects on pain and pain-related dysfunction were similar. Nevertheless, heat massage was shown to have a better effect on autonomic nerve function and the underlying mood.

The limitations of this study included the small sample size, short study period, and exclusive focus on subacute LBP. For the purposes of generalizability, a comparative study is warranted in the future, which includes an adequately powered sample size and a longer duration of treatment application with other therapeutic modalities.

## 5. Conclusions

The resultant effects on pain and pain-related dysfunction were shown to be similar in this study that compared heat massage with conventional physical therapy. Both treatments were shown to be effective in improving pain and pain-related disability. Combination heat massage management is not superior to conventional physical therapy. Nevertheless, heat massage was shown to have a better effect on the control of autonomic nerve function and underlying mood.

## Figures and Tables

**Figure 1 healthcare-11-00991-f001:**
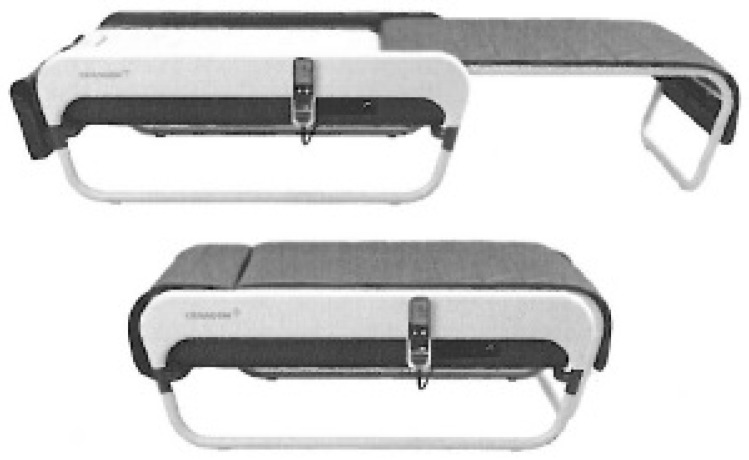
The heat massage device (CGM MB-1401; Ceragem, Republic of Korea).

**Figure 2 healthcare-11-00991-f002:**
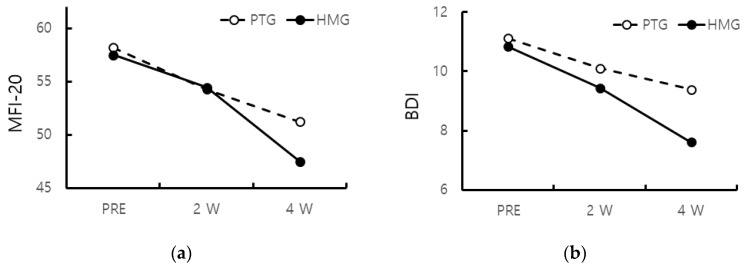
(**a**) MFI-20 of both groups were improved from 2 W, and the MFI-20 of the HMG improved from 4 W compared to that of the PTG (*p* < 0.05); (**b**) BDI of the PTG was improved at 4 W compared to PRE and 2 W, but that of the HMG was improved at 2 W (*p* < 0.05). (MFI-20: multidimensional fatigue inventory 20, BDI: Beck’s depression inventory).

**Figure 3 healthcare-11-00991-f003:**
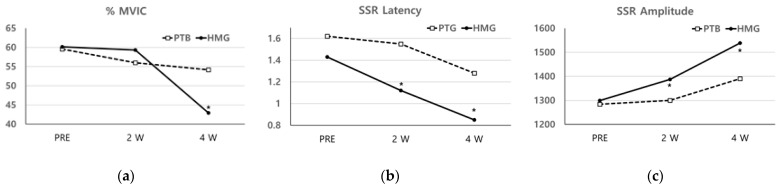
(**a**) The levels of the normalized sEMG signals (%MVIC) were significantly decreased at 4 W in both groups. However, there was a decrease in the HMG compared to the PTG at 4 W (*p* < 0.05); (**b**,**c**) in a comparison between the two groups, the latency, and amplitude of the SSR were significantly different at 2 W and 4 W compared to at PRE (*p* < 0.05). (%MVIC: the level of the normalized sEMG signal, SSR: sympathetic skin response).

**Table 1 healthcare-11-00991-t001:** Characteristics of the subjects.

Characteristics	Categories	Exp.(*n* = 20)	Cont.(*n* = 20)	χ^2^ or t	*p*
		*n* (%) or M ± SD	*n* (%) or M ± SD		
Sex	Male	9 (45.0)	10 (50.0)	0.100 ^‡^	0.752
	Female	11 (55.0)	10 (50.0)		
Age		56.30 ± 12.28	56.15 ± 9.75	0.043 ^†^	0.966
	50>	3 (15.0)	2 (10.0)	0.259 ^‡^	0.879
	50~60	8 (40.0)	9 (45.0)		
	60<	9 (45.0)	9 (45.0)		
Onset (month)		12.10 ± 2.73	11.85 ± 2.48	0.303 ^†^	0.763

Values are presented as the mean ± SD deviation. Exp. = experimental group; Cont. = control group. ^†^ Independent sample *t*-test. ^‡^ Chi-squared test.

**Table 2 healthcare-11-00991-t002:** Homogeneity test for the a priori dependent variables.

Variables	Exp. (*n* = 20)	Cont. (*n* = 20)	Z ^†^	*p*
	M ± SD	M ± SD		
BMI	25.83 ± 2.62	25.93 ± 4.10	−0.406	0.685
Back rest	42.64 ± 20.66	41.21 ± 24.17	−0.271	0.787
%MVIC	60.42 ± 19.57	59.86 ± 18.87	−0.054	0.957
Lt_Lat	1298.90 ± 106.82	1283.80 ± 92.18	−0.379	0.705
Lt_Amp	1.49 ± 0.43	1.62 ± 0.50	−0.758	0.448
MPQ(Sensory)	7.65 ± 1.14	7.90 ± 0.85	−0.650	0.516
MPQ(Affective)	1.35 ± 0.67	1.40 ± 0.68	−0.270	0.787
SF-MPQ	9.10 ± 1.74	9.30 ± 1.22	−0.249	0.803
BDI	11.05 ± 3.56	11.10 ± 3.01	−0.041	0.967
MFI	57.75 ± 2.10	58.15 ± 2.06	−0.746	0.456
VAS	5.65 ± 1.14	5.80 ± 1.15	−0.267	0.790
ODI	29.22 ± 5.36	29.33 ± 6.56	−0.222	0.825
RMDQ	5.70 ± 1.38	5.95 ± 1.32	−0.519	0.604
Epi	68.52 ± 58.71	59.78 ± 17.14	−0.850	0.395
NE	286.47 ± 97.37	328.24 ± 128.90	−0.394	0.694
Cortisol	7.30 ± 2.80	6.34 ± 3.79	−1.120	0.263

Exp. = experimental group; Cont.= control group. ^†^ Mann–Whitney U-test.

**Table 3 healthcare-11-00991-t003:** Comparison of the blood sample tests.

	HMG	PTG	*p*-Value
AST (aspartate aminotransferase)			
Baseline (PRE)	24.44 (1.48)	25.75 (0.94)	0.495
4 weeks	23.88 (0.99)	25.81 (1.06)	0.068
LDH (lactate dehydrogenase)			
Baseline (PRE)	177.63 (7.41)	184.06 (7.69)	0.551
4 weeks	177.69 (8.43)	194.69 (11.97)	0.255
Lactate			
Baseline (PRE)	0.95 (0.08)	1.31 (0.15)	0.057
4 weeks	1.21 (0.07)	1.51 (0.23)	0.219
Creatine kinase			
Baseline (PRE)	110.75 (11.11)	132.19 (25.43)	0.446
4 weeks	112.31 (13.48)	130.88 (26.53)	0.538
Epinephrine			
Baseline (PRE)	66.66 (14.30)	60.37 (4.18)	0.679
4 weeks	47.55 (5.74) *	48.58 (4.31)	0.887
Norepinephrine			
Baseline (PRE)	285.86 (23.52)	343.02 (34.46)	0.181
4 weeks	234.95 (20.27) *	309.44 (37.92)	0.072
Cortisol			
Baseline (PRE)	7.19 (0.68)	6.35 (0.91)	0.463
4 weeks	7.33 (0.74)	6.10 (0.77)	0.263

Values are presented as the mean (SD), * Mann–Whitney U-test, *p* < 0.05.

**Table 4 healthcare-11-00991-t004:** Comparison of the pain and pain-related disabilities.

Dependent Valuables		Baseline (PRE)	2 Weeks (2 W)	4 Weeks (4 W)	χ^2 ‡^	Within-Group Comparisons
PNRS	HMG	5.50 (1.10)	4.67 (1.19)	3.17 (1.29)	35.690	PRE > 2 W, 4 W
	PTG	5.78 (1.22)	4.94 (1.26)	3.39 (1.46)	33.549	PRE > 2 W, 4 W
	Z ^†^	−0.267	−0.057	−0.252		
	*p*	0.790	0.955	0.801		
ODI	HMG	29.26 (5.67)	21.60 (3.56)	11.73 (5.43)	37.026	PRE > 2 W, 4 W
	PTG	29.38 (6.93)	20.00 (7.43)	11.36 (5.44)	32.316	PRE > 2 W, 4 W
	Z ^†^	−0.222	−0.689	−0.141		
	*p*	0.825	0.491	0.888		
RMDQ	HMG	5.61 (1.42)	3.78 (1.17)	2.50 (0.51)	34.105	PRE > 2 W, 4 W
	PTG	6.00 (1.37)	4.06 (1.16)	2.72 (0.89)	30.907	PRE > 2 W, 4 W
	Z ^†^	−0.519	−0.397	−0.151		
	*p*	0.604	0.691	0.880		

Values are presented as the mean (SD), PNRS: pain numeric rating scale, ODI: Oswestry disability index, RMDQ: Roland–Morris disability questionnaire, ^†^ Mann–Whitney U-test, ^‡^ Friedman test, *p* < 0.05.

**Table 5 healthcare-11-00991-t005:** Comparison of the SF-MPQ Parameters.

Dependent Valuables		Baseline (PRE)	2 Weeks (2 W)	4 Weeks (4 W)	χ^2 ‡^	Within-Group Comparisons
SF-MPQ (Sensory)	HMG	7.67 (1.19)	6.00 (0.97)	4.17 (1.29)	30.737	PRE > 2 W, 4 W
	PTG	8.00 (0.84)	5.94 (1.00)	4.11 (0.83)	38.519	PRE > 2 W, 4 W
	Z ^†^	−0.650	−0.058	−0.014		
	*p*	0.516	0.954	0.989		
SF-MPQ (Affective)	HMG	1.33 (0.69)	0.83 (0.38)	0.56 (0.51)	20.440	PRE > 2 W, 4 W
	PTG	1.39 (0.70)	0.94 (0.24)	0.72 (0.46)	16.625	PRE > 2 W, 4 W
	Z ^†^	−0.270	−0.874	−0.637		
	*p*	0.787	0.382	0.524		
SF-MPQ	HMG	9.00 (1.78)	6.83 (1.10)	4.72 (1.93)	30.727	PRE > 2 W, 4 W
	PTG	9.39 (1.24)	6.89 (1.02)	4.83 (0.92)	38.519	PRE > 2 W, 4 W
	Z ^†^	−0.249	−0.346	−0.304		
	*p*	0.803	0.730	0.761		

Values are presented as the mean (SD), SF-MPQ: short-form McGill pain questionnaire, ^†^ Mann–Whitney U-test, ^‡^ Friedman test, *p* < 0.05.

## Data Availability

Data will be available upon reasonable request to the authors.

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
