# Peer review of "Substantiating the Therapeutic Effects of Simultaneous Heat Massage Combined with Conventional Physical Therapy for Treatment of Lower Back Pain: A Randomized Controlled Feasibility Trial"

_healthcare, 2023, doi:10.3390/healthcare11070991_

Round 1

Reviewer 1 Report

Thank you for this interesting paper Please take a look at the attached word document..

Reviewer 2 Report

Introduction

Lines 43-53: please check referencing style as it is not consistent

Line 44: "...have reported that LBP compromise..." check grammar

Line 45: "The functionality of individuals  are profoundly impacted" check grammar, change are to is.

Line 46-48: please rephrase as sentence is not making sense

Line 60: "and low frequency massage" please elaborate. You mean not often during the week or done in a slow manner?

Lines 66-70: you refer to a specific technique? this is a gneralisation of massage and the reference [6] is not adequate to support this

Lines 75-76: "The clinical effects of combined heat and massage application are increased blood 75 flow to affected areas, reduced muscle spasms, and increased flexibility" this statement needs referencing

Materials and Methods

Line 90: "This pilot study is a double-blinded" as the physios knew what treatment they were administering, it would not have been a double but a single blinded study. If you believe otherwise, please explain.

Line 91: "by our Institutional Review Board" you have to be specific which

Lines 94-96: "We enrolled 20 participants in each of the two groups (N=40) by convenience sampling and exceeds the minimum recommended number of participants for pilot studies"  where did you get your participants? from a clinic? from university students via a general call? please elaborate on convenience sampling.

Line 102:  "The study was conducted by a third-party clinical research coordinator" please explain this statement further . The writers of this paper were not involved in any way?

Lines 128-129: "All participants used the device for 40 minutes, once daily, 5 days per week for 4 total weeks." why did you choose this treatment dosage? where there any specific citeria? 

Lines 133-135: "A typical physiotherapy program without 133 exercises applied 5 minutes of therapeutic ultrasound (US), 10 minutes of transcutaneous 134 electrical nerve stimulation (TENS), and 15 minutes of hot pack application." please specify the area of application and U/S and TENS settings.

Line 138: "Participants who received physical therapy or simultaneous heat-massage" was there a third group that received both intervetions? 

Line 154: it should write short-form not short-term

Line 167: "recording electrodes was attached at the L3 vertebral level," please explain how you defined the location. Palpation? Accuracy?

Line 216: "and these values were not 216 significant difference" check grammar

Results

Line 241: "The MFI-20 was changed to" you mean from baseline (pre) to the 2W and 2W. please consider rephrasing to make more sense. same for the next sentence and the following. also check the grammar of this whole section (3.4)

Line 253: "The normalization of sEMG signal..." please bear in mind that sEMG normalisation is a process and the process did not change. The levels of the normalised sEMF values changed. Consider revising

Discussion

Lines 273-274: "patients 273 who has had" check grammar

Line 275: this sub heading seem to refer to the whole of the discussion that follows. consider removing

Line 329: " is defined by reducing..." please check grammar

Author Response

Thank you for your comments.
Please see the attached Word file.

Round 2

Reviewer 2 Report

I am happy with the changes. However, the script needs to be checked again for grammar and syntax errors

Author Response

Thank you for your comments.
I revised the overall grammar and syntax errors through MDPI English editing.
